# Predictive metabolomic profiling of microbial communities using amplicon or metagenomic sequences

Himel Mallick [1,2], Eric A. Franzosa [1,2], Lauren J. McIver[1,2], Soumya Banerjee [1,2], Alexandra Sirota-Madi[1,2], Aleksandar D. Kostic [1,2], Clary B. Clish [1], Hera Vlamakis[1], Ramnik J. Xavier [1,3,4,5] & Curtis Huttenhower [1,2]

Microbial community metabolomics, particularly in the human gut, are beginning to provide a new route to identify functions and ecology disrupted in disease. However, these data can be costly and difficult to obtain at scale, while amplicon or shotgun metagenomic sequencing data are readily available for populations of many thousands. Here, we describe a computational approach to predict potentially unobserved metabolites in new microbial communities, given a model trained on paired metabolomes and metagenomes from the environment of interest. Focusing on two independent human gut microbiome datasets, we demonstrate that our framework successfully recovers community metabolic trends for more than 50% of associated metabolites. Similar accuracy is maintained using amplicon profiles of coral-associated, murine gut, and human vaginal microbiomes. We also provide an expected performance score to guide application of the model in new samples. Our results thus demonstrate that this 'predictive metabolomic' approach can aid in experimental design and provide useful insights into the thousands of community profiles for which only metagenomes are currently available.

---

[1] Broad Institute of MIT and Harvard, Cambridge, MA 02142, USA. [2] Department of Biostatistics, Harvard T. H. Chan School of Public Health, Boston, MA 02115, USA. [3] Center for Computational and Integrative Biology, Massachusetts General Hospital and Harvard Medical School, Boston, MA 02114, USA. [4] Gastrointestinal Unit and Center for the Study of Inflammatory Bowel Disease, Massachusetts General Hospital and Harvard Medical School, Boston, MA 02114, USA. [5] Center for Microbiome Informatics and Therapeutics, Massachusetts Institute of Technology, Cambridge, MA 02139, USA. Correspondence and requests for materials should be addressed to R.J.X. (email: xavier@molbio.mgh.harvard.edu) or to C.H. (email: chuttenh@hsph.harvard.edu)

Advances in high-throughput metabolomics technology have enabled comprehensive coverage of a large number of small-molecule metabolites in microbial communities[1]. Analysing metabolic differences between differentially regulated biochemical pathways can facilitate the discovery of potential biomarkers associated with disease and provide insights into the underlying pathogenesis[2,3]. This has been highlighted by an increase in studies that rely on multi'omic profiling to simultaneously characterize community ecology, metabolic signatures, and functional attributes of the human microbiome or other environments[4–12]. For example, among the best-studied bioactive microbial metabolites influencing human health are the short-chain fatty acids (SCFAs) including propionate, butyrate, and acetate, which have been implicated in the pathogenesis of several diseases, including inflammatory bowel disease (IBD) and colorectal cancer[13–15]. Other examples include the bile acids[16], sphingolipids[17], and tryptophan derivatives[18] all with evidence of microbial interactions and bioactivity in the gut.

Inferring the capacity of a microbial community to produce molecules and using large-scale data sets to connect new specific genes to metabolites is thus an essential first step towards the goal of understanding how and why gut microbiome metabolism affects human health[19]. The strength of association between gut microbial and metabolic profiles suggests that it may be possible to approximately predict the metabolomic activities or features of microbial communities from metagenomes, based on their taxonomic or functional profiles. Easily identifying such associations purely based on enzymatic roles is greatly limited by the currently unsaturated repertoire of gene–metabolite reactions, as well as by the relative (rather than absolute) abundance measures provided both by typical sequencing and metabolomic technologies. Despite these limitations, however, approaches that predict metabolite features associated with gut microbial profiles can serve as a hypothesis generator that can facilitate population-scale discovery of novel associations (e.g. in large metagenomic data collections) and lead to new sets of testable hypotheses, serving as a complementary adjunct to experimental validation studies (e.g. as has been the case for predictive functional profiling from amplicon data[20]).

Recently, a few studies have taken initial steps to carry out such predictions in the subset of cases with prior knowledge of the mechanisms linking microbiome and metabolome (e.g. from stoichiometric enzyme reaction matrices derived from databases such as Kyoto Encyclopedia of Genes and Genomes (KEGG)[21]). One set of approaches, collectively referred to as Predicted Reactive Metabolic Turnover (PRMT), calculate community-based metabolite potential (CMP) scores, which represent the relative capacity of the community in a given sample to generate or deplete each metabolite[22–24]. Other methods reconstruct predictive metabolic models of community metabolism in either a constraint-based or a network-based modeling framework[25–27]. One common drawback of both these approaches is their inability to distinguish between failure to predict due to missing annotation or accurate reaction information in the reference database and failure due to alternative biological mechanisms, making them difficult to apply or validate in a data-driven manner. In addition, these methods depend on accurate characterization and annotation of species- and even strain-specific metabolites, and they do not scale well to complex communities with partially referenced taxa or metabolites. All these studies thus link microbial functional potential to metabolomic activity, but they are limited in scope and lack rigorous external (independent) validation, particularly in environments such as the human gut in which metabolomic training measurements are feasible and where accurate and novel bioactive discovery can have particular impact.

Here we describe MelonnPan (Model-based Genomically Informed High-dimensional Predictor of Microbial Community Metabolic Profiles), a computational framework to predict community metabolomes from microbial community profiles. MelonnPan infers the composite metabolome by enabling (1) data-driven identification of an optimal set of predictive microbial features, and (2) robust quantification of the prediction accuracy of the well-predicted metabolites. This allows researchers to reproducibly infer metabolites for communities from which only metagenomes are currently available. We applied MelonnPan to two independent gut metagenome data sets comprising >200 patients with Crohn's disease (CD), ulcerative colitis (UC), and healthy control (HC) participants. This revealed high concordance between predicted and observed community metabolic trends in >50% of metabolites whose identities were confirmed against laboratory standards, including prediction of metabolic shifts associated with bile acids, fatty acids, steroids, prenol lipids, and sphingolipids. When using taxonomic features from amplicon sequencing profiles, similar accuracy was maintained for coral-associated, murine gut, and human vaginal microbial communities as well. The implementation of MelonnPan, associated documentation, and example datasets are made freely available in the MelonnPan software package at http://huttenhower.sph.harvard.edu/melonnpan.

## Results

**The MelonnPan algorithm.** We have developed MelonnPan as a computational method to predict metabolite features from amplicon or metagenomic sequencing data by incorporating biological knowledge in the form of either taxonomic or functional profiles. Unlike existing stoichiometry-based methods that rely on a limited number of well-characterized taxa, enzymes, and metabolites, functional annotation is not necessary for MelonnPan, as the tool is designed to capture insights using machine learning even from uncharacterized microbial features. In this manuscript, we discuss specifically its application to the human gut microbiome, but the methodology is generalizable to any appropriately profiled microbial environment. Briefly, MelonnPan uses elastic net regularization[28] to identify which features (taxonomic or functional) are predictive for a given metabolite. Given a new taxonomic profile (from amplicons or a metagenome) or metagenomic functional profile (i.e. gene family abundances), it then combines a subset of the sequence features to estimate the associated composite metabolome. The resulting predicted metabolites are each the weighted sum of relative abundances of predictive features (taxa or gene families), where the regression coefficients from the trained elastic net model are used as weights in the prediction algorithm (Fig. 1).

In the fitting stage, MelonnPan is trained using samples for which both sequencing data and experimentally measured metabolite abundances are available (Fig. 1a). Both measures are effectively relative abundances—normalized reads or spectral counts, respectively. The training (and, later, inference) metagenomes can be profiled with any system that quantifies relative abundances of taxa or functionally related microbial gene families; here we use previously profiled amplicon data and metagenomes functionally profiled by HUMAnN2[29] with UniRef90 as the reference catalogue, i.e. clustered sets of sequences from UniProt at a minimum of 90% amino acid identity[30]. The fitting process uses per-metabolite elastic net regularization to optimize a small number of sequence features' coefficients. The final model for a particular environment (the human gut or otherwise) is selected based on rigorous internal validation (cross-validation) corresponding to the greatest cross-validated likelihood for each metabolite. Metabolites that cannot be well

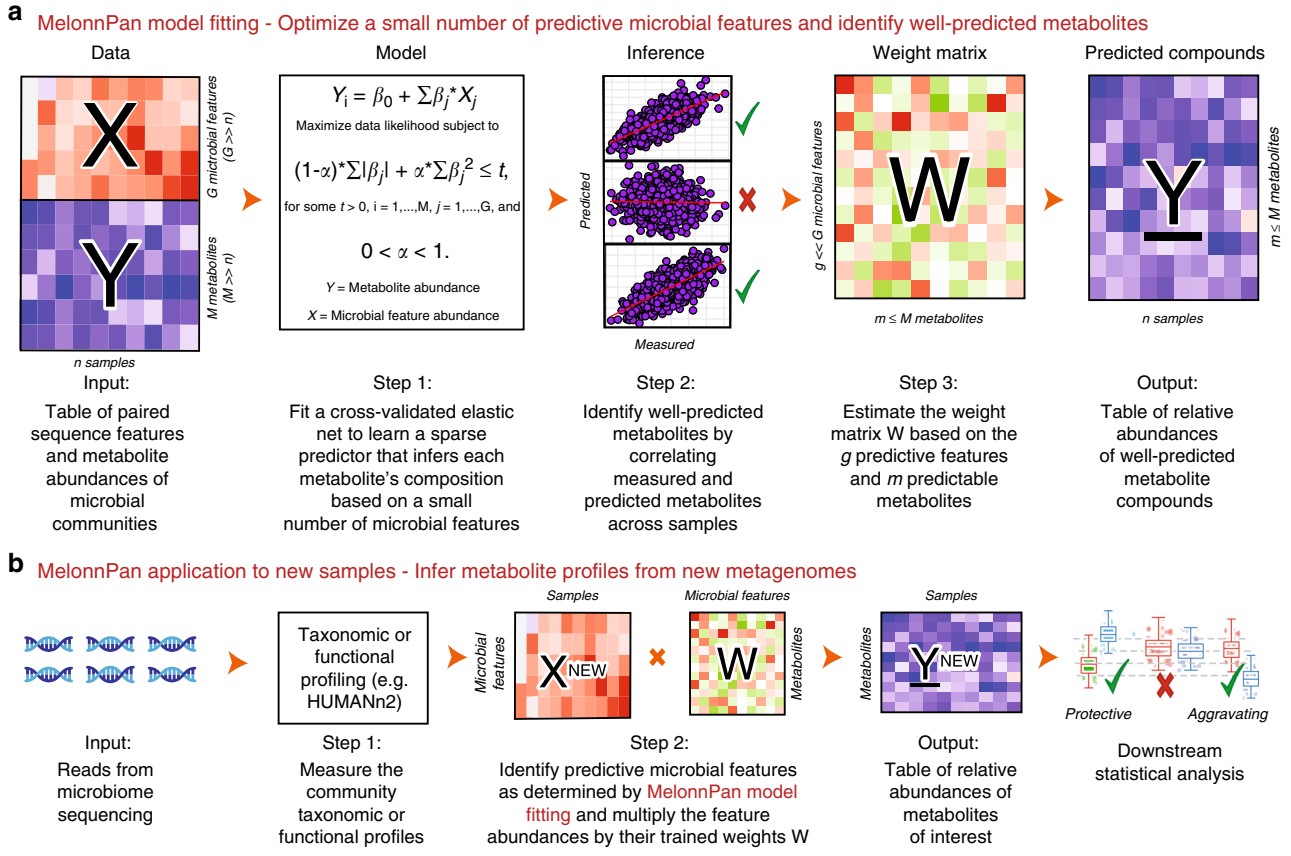

**Fig. 1** MelonnPan is a predictive model inferring microbial community metabolite features from amplicons or metagenomes. **a** The MelonnPan model can be trained to infer metabolite profiles for a particular microbial community type given, first, training data consisting of paired metagenomes (**X**) and metabolomes (**Y**) from the environment of interest. The model is fit beginning with microbial sequence features derived from training metagenomes. It uses an elastic net regularized regression, per metabolite, to identify a minimal set of microbial features whose abundances predict that metabolite. These individual learners are first checked using cross-validation, and poorly fit metabolites (Spearman correlation coefficient between measured and predicted metabolite abundances across samples <0.3) are flagged. **b** The sequence features' coefficients (**W**) for remaining, well-predicted metabolites are saved and can be applied to new metagenomes to predict the associated metabolite features (**Y**, in units of relative abundance), which can be utilized for downstream epidemiological analysis

predicted by any generalizable model are flagged based on rank correlation between predictions and training measurements (Spearman correlation coefficient <0.3). Finally, the model can then be applied to new microbial communities from analogous environments using simple linear regression, multiplying the learned coefficient values by sequence feature abundances (Fig. 1b). During model assessment prior to predictive applications in new data sets, no information from the test set is used in training the model. Here, for our main gut-specific model, we have also applied the final internally validated model to an independent, external validation cohort. Performance is summarized as each metabolite's Spearman's rank correlation coefficient across all samples with the corresponding measured metabolite ("Methods").

**MelonnPan accurately predicts metabolites from metagenomes.** We validated an initial MelonnPan model for the human gut using two independent metagenomic and metabolomic data sets comprising 155 and 65 IBD patients and controls, respectively, with CD ($n = 68$ and 20), UC ($n = 53$ and 23), and HC participants ($n = 34$ and 22; Supplementary Table 1). In each cross-sectional cohort (the Prospective Registry for IBD Studies at the Massachusetts General Hospital (PRISM) and the Netherlands IBD cohort (NLIBD)), stool samples were profiled by a combination of shotgun metagenomic sequencing and four liquid

chromatography tandem mass spectrometry (LC-MS) methods (including polar compounds in the positive and negative ion modes, lipids, and free fatty acids and bile acids, "Methods"). The LC-MS profiling yielded ~8000 clustered features, characterized by chromatographic retention time and exact mass. Metagenomic functional profiles were generated for all samples with HUMAnN2[29], resulting in approximately 1 million UniRef90 gene families. Both data types were quality controlled and filtered before modeling: features were removed when they did not vary in value over the available samples. In particular, both gene families and metabolites, of very low relative abundance and prevalence (<0.01% in ≥10% of samples) were removed, leaving 2818 metabolites and 814 gene families for final modelling ("Methods"). All training, including internal cross-validation, was performed using only data from PRISM; performance evaluation and external validation was conducted using held-out samples from the Netherlands (NLIBD).

In these data, we modelled a panel of 466 metabolites whose identities were confirmed experimentally against laboratory standards[31]. After initial filtering, model fitting, and internal cross-validation, >50% ($n = 107$, 53.8%) were well predicted (Spearman correlation of predicted versus measured metabolite abundance ≥0.3) by MelonnPan during independent validation (Supplementary Fig. 1). The well-predicted metabolites (Fig. 2a) included sphingolipids [e.g. ceramide and phytosphingosine, fatty

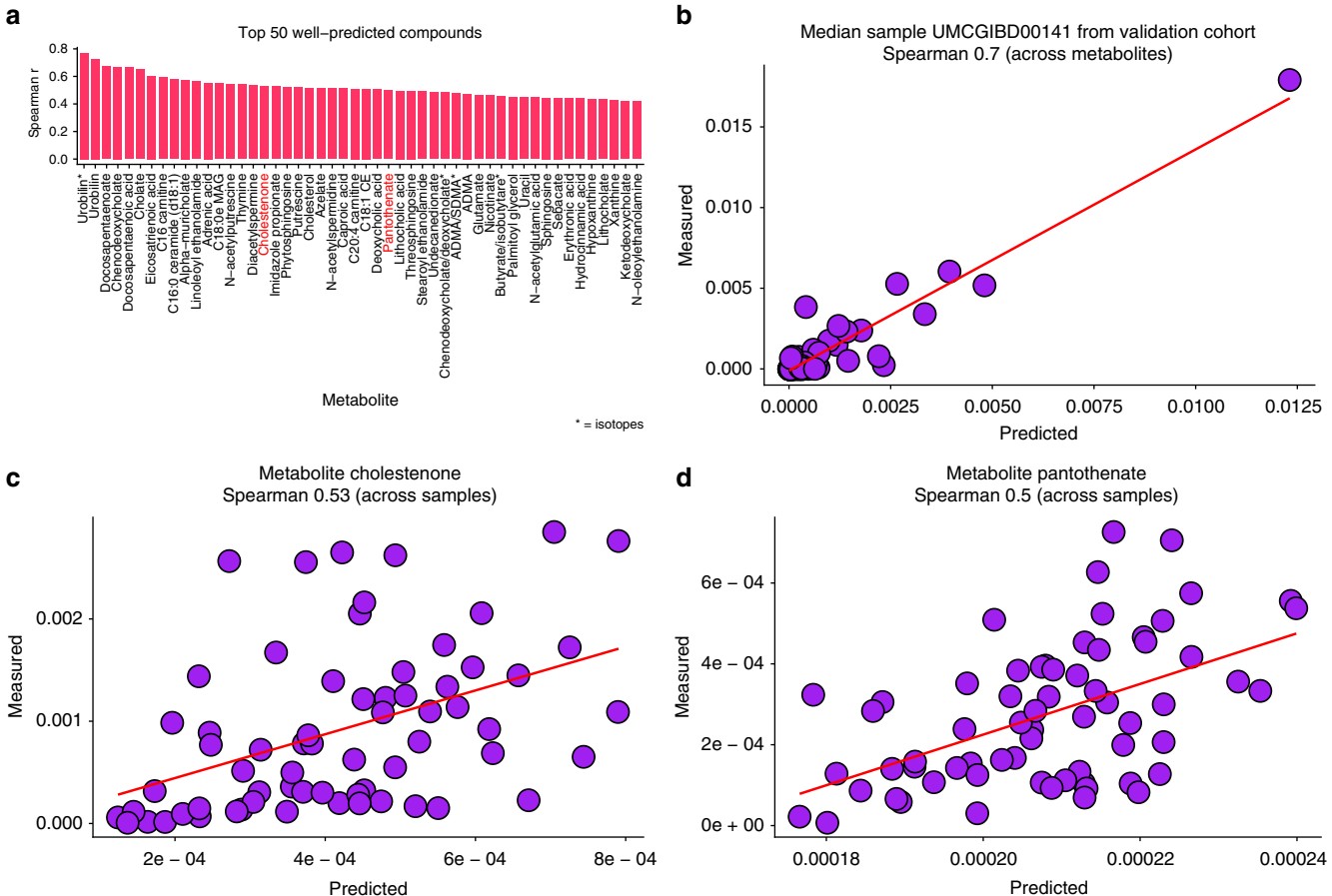

**Fig. 2** MelonnPan accurately predicts metabolite features based on metagenomic sequence profiles. **a** From a panel of 466 metabolites whose identities were confirmed against laboratory standards, these 50 were the best predicted by MelonnPan with unique Human Metabolome Database (HMDB) identifiers (as measured by Spearman correlation ($r$) of predicted versus measured profiles in the Netherlands inflammatory bowel disease (NLIBD) independent validation cohort). All metabolites shown have $r > 0.3$ over a total $n = 65$ NLIBD samples. Two representative metabolites are shown in red. **b** Measured and predicted metabolite profiles across a single representative sample were strongly and significantly associated across 107 labelled metabolites (including both unique and non-unique compounds that were well predicted by MelonnPan). **c** Representative significant prediction for cholestenone in the test set. **d** Representative statistically significant prediction for pantothenate in the NLIBD validation data (values are relative abundances). See Supplementary Figs. 1–2 and Supplementary Data 1–6 for full results. For each scatter plot, the best fitting regression line is also shown (in red)

acids (e.g. docosapentaenoic acid and caproic acid), B-group vitamins (e.g. pantothenate, Fig. 2d), and derivatives of cholesterol and bile acids [e.g. cholestenone (Fig. 2c) and cholic acid]. These compounds are increasingly recognized as important signalling molecules in the regulation of systemic host–microbial cometabolism[1]. The fully fit model parameters, predictions, and performance summary across metabolites and subjects are provided in Supplementary Data 1–4, and the implementation and model are available online at http://huttenhower.sph.harvard.edu/melonnpan.

A substantially large number of metabolites were thus well predicted across samples (Supplementary Data 5), both during cross-validation and in independent held-out metagenomes; as a result, MelonnPan can successfully recover the metabolomic profiles of a moderately large number of experimentally validated metabolites across entire metagenomic samples (Fig. 2b, Supplementary Data 6, Supplementary Fig. 2). While these validations mainly tested the subset of identified metabolites verified using experimental standards, the model can be trained and fit to all unique metabolites, even those without a confidently assigned label (Supplementary Fig. 3). In particular, among 2818 metabolites with unique cluster IDs, approximately 60% ($n =$

1679, 59.6%) had ≥0.3 accuracy (Spearman $r$) during training (Supplementary Fig. 1). Among these, a large number ($n = 933$, 55.6%) were, like the labelled compounds, well predicted both during cross-validation and in independent validation data. This high predictability is somewhat surprising, given the complexity of gut microbial communities and the multitude of external and internal influences that can potentially impact metabolite abundances. This highlights the robustness of MelonnPan's predictive capabilities even in the context of complex, unstable, and dynamic communities, such as the human gut.

In each well-predicted metabolite, on average, <2% of the gene families (median model size = 12, median positive weights = 9, and median negative weights = 11) were selected by the MelonnPan model (Supplementary Data 3, Supplementary Fig. 4). This is both statistically and biologically meaningful, since MelonnPan imposes regularization to estimate a sparse model and identify a small set of relevant sequence features (where both the elastic net mixing and sparsity parameters are selected based on internal ten-fold cross-validation; "Methods"). The learned weights from MelonnPan represent the relative capacity of the features in a given sample to be associated (positively or negatively) with each metabolite (assuming some baseline metabolite profile across

samples). Biologically, this allows us to build an interpretable model by progressively setting the contributions of less relevant features to zero and retaining only a small number of features—in this case, genes with potential enzymatic contributions to, or other associations with, the targeted metabolite.

As an additional validation, and to further rule out the possibility of data artefacts, we tested MelonnPan's behaviour when attempting to link randomized, null microbial profiles to metabolite profiles. In particular, we independently permuted both metabolite and gene family training data across samples, which were then renormalized following permutation to preserve the core characteristics of each individual data set ("Methods"). We repeated this procedure 1000 times, each time collecting the resulting coefficients from the trained MelonnPan model, and averaged the number of well-predicted metabolites across iterations to derive the final predictability (which would, in the absence of overfitting, remain near-zero). We found that the randomized null profiles produced a consistently very low set of metabolites considered to be well predicted as compared to true, unshuffled data during assessments on both training and independent validation data sets (59.6% of true compounds well predicted versus 3.2% after permutation of training data; 55.6% of true compounds well predicted in validation data, versus 4.4% after permutation, McNemar's exact test $P < 0.0001$, Supplementary Figs. 5–6). This represents both an indication of MelonnPan's robustness to overfitting and a justification for its arbitrary threshold (i.e. Spearman $r > 0.3$), which is broadly consistent for summarizing a sufficiently high number of "well-predicted" metabolites.

**Estimating MelonnPan accuracy in new microbial communities**. The usefulness of MelonnPan, of course, depends on the accuracy of its predicted metabolomes from new microbial community samples and the corresponding ability to recapitulate findings from metabolomic studies. To characterize this effect, we developed the Representative Training Sample Index (RTSI) (in the spirit of the Nearest Sequence Taxon Index of PICRUSt[20]) to quantify the representativeness of new samples with respect to training data sets ("Methods"). Briefly, MelonnPan first flags any feature (taxon or gene family) not present in the training metagenomes, and for the remaining common features (between training and test samples), it calculates an average similarity score (per microbial community sample) based on principal component analysis (PCA). In particular, RTSI scores are computed by sequentially seeking the highest correlation coefficient with the top principal components (PCs) explaining a majority of the variation in the training microbiomes ("Methods"). We compared the RTSI scores and MelonnPan accuracies for all of the NLIBD validation samples across all well-predicted metabolites (Supplementary Fig. 7), which revealed a strong and statistically significant correlation (Spearman correlation = 0.4, $P = 0.003$) between representativeness (higher RTSI) and predictability of samples across metabolites (as measured by Spearman correlation between measured and predicted metabolite abundances). This correlation value was itself conservatively low, caused by a few outlier samples without which the predictiveness of the RTSI for MelonnPan performance on new samples is even greater.

These insights have potential implications at multiple levels. First, this method provides MelonnPan users with a way to be appropriately cautious when applying the model to predict metabolite features for communities distinct from a default, human gut model or from other models that the user may have trained. The ability to calculate RTSI values within MelonnPan allows users the flexibility to determine whether their samples

are similar enough for a trustworthy MelonnPan prediction before running an analysis. Second, these evaluation results in the NLIBD cohort confirm that MelonnPan predictions can be conceived as surrogates for metabolic profiles of communities across a large number of representative metagenomes, in the absence of measured metabolomic data. This can serve both as a hypothesis-generation tool and as an efficient, cost-effective first pass analysis driving experimental design, which we recommend pairing with follow-up experiments to prove the inferred metabolite profiles (much as has been the case for amplicon and metagenomic data analysis using PICRUSt[20]).

**MelonnPan predictions outperform existing methods**. We next sought to compare our results with the predicted metabolites identified by a recently developed metabolite prediction method, MIMOSA[24]. MIMOSA was evaluated, as it is, to the best of our knowledge, the only current method capable of predicting community-wide metabolic relative abundances from population-level metagenomic data, as well as providing an informed software implementation. MIMOSA builds on a previously proposed metabolic network model (PRMT[22]) to estimate the metabolic potential of a microbial community from taxonomic composition and metagenome content. Briefly, MIMOSA first generates a stoichiometric matrix describing the quantitative relationship between genes and metabolites to provide an estimate of the CMP score of the community of interest. Next, it compares the differences in CMP scores between all pairs of samples with the differences in the corresponding measured metabolites. In order to identify statistically significant well-predicted metabolites, MIMOSA relies on false discovery rate (FDR)-corrected $P$ values based on Mantel's test[32] for correlation between two distance matrices. Similar to MelonnPan, MIMOSA also relies on user-provided paired tables of metabolites and microbial sequence features. However, unlike MelonnPan, it does not explicitly use data mining and model building to construct and validate a predictive model. As a result, MIMOSA is unable to do prediction on new metagenomic samples not previously seen by the algorithm.

In order to assess the prediction performance of MIMOSA in the NLIBD cohort, we first mapped the corresponding metabolite names to KEGG identifiers by mapping the compound IDs to Human Metabolome Database (HMDB[33]), which includes cross-references to KEGG compound identifiers, leading to 303 KEGG compounds (149 unique identifiers). Next, we converted the UniRef90 gene family abundances to approximate the corresponding KO (KEGG Orthology) abundances by assigning the UniProt-KO annotations to the corresponding protein families in UniRef90. In order to apply MIMOSA, we first normalized the KO abundances using MUSiCC[34] (default and recommended option in the MIMOSA software). Only a small number of metabolite compounds were well predicted by MIMOSA (Supplementary Fig. 8A; $n = 20$ (23%), Mantel's test $Q < 0.05$). In contrast, MelonnPan was able to accurately predict the vast majority of these metabolites ($n = 130$ (84%), Spearman $r > 0.3$). In addition, a few metabolites were anti-predicted by MIMOSA, unlike MelonnPan which generally yielded higher confidence (greater Spearman correlation between measured and predicted abundances) among the common metabolites predicted by both methods (Supplementary Fig. 8B) and greater number of well-predicted metabolites even when restricted to the small subset of well-characterized metabolites (Supplementary Fig. 8C). This suggests that there may be major gaps between the relatively small proportion of annotated microbial enzymatic activities and those newly identifiable using machine learning.

**MelonnPan uncovers meaningful biological relationships.** To gain insights into the taxonomic and functional makeup of the most predictive gene families in this context, we next quantified how much each gene family contributed to the MelonnPan predictions for the human gut. We performed gene set enrichment analysis (GSEA)[35] to identify the classes of genes most frequently selected by the metabolic model, i.e. significantly over- or under-represented during metabolite prediction (ranking gene family features by how often they are used in a model for any well-predicted metabolite). We assigned taxonomy to these gene families using HUMAnN2[29] by choosing the lowest common ancestor of the majority of genes with homology, based on the gene's taxonomic assignment (or "Unclassified" when not unique to the genus level). We then compared the general difference in the cumulative distributions of the gene families in each gene set (summarized to genera) with that of the MelonnPan ranked list with a permutation-based Kolmogorov–Smirnov (KS) test ("Methods").

Eight genera were significantly over-abundant in the MelonnPan gene list, with the strongest effects observed among *Pseudoflavonifractor*, *Clostridium*, *Coprococcus*, *Anaerotruncus*, *Blautia*, *Collinsella*, *Ruminococcus*, and *Anaerostipes* (statistically significant GSEA result with $Q < 0.25$, Fig. 3a, Supplementary Fig. 9). The majority of these genera are from the Firmicutes phylum, with the exception of *Collinsella* which belongs to the phylum Actinobacteria. Some of these genera including *Clostridium* and *Ruminococcus* encode several species belonging to the *Clostridium* cluster IV or XIVa[36] that preferentially colonize the mucus layer and consequently increase the butyrate bioavailability for colon epithelial cells[37]. A decrease in the relative abundances of these species in the human colon has been associated with several diseases, including IBD[38]. Moreover, species from *Clostridium* cluster IV are also known to be the primary producers of SCFAs in the human colon, which are increasingly recognized as key signalling molecules between the gut microbiota and the host[13]. Decomposition of these genera revealed that they were further typically contributed by a few representative species or strains (Supplementary Data 7). MelonnPan thus enables identification of functionally relevant gene families with species- or even strain-specific metabolic associations, facilitating biologically relevant mechanistic studies at finer taxonomic resolution.

In order to further decipher these gene families at greater resolution, we repeated the enrichment analysis using functional annotations to identify biological processes that were significantly over- or under-abundant in metabolite prediction ("Methods"). We focussed on the Pfam database[39], which categorizes these metabolically predictive gene families into protein families. Surprisingly, while no individual Pfam families were enriched during this testing, there was a significant over-representation of uncharacterized protein domains among predictive gene families (Fisher's exact test $P = 3.46\text{e}-52$, Fig. 3b), which was consistent across all functional annotation catalogues we considered (Supplementary Fig. 10). This substantially large number of unannotated genes likely include as yet uncharacterized metabolic enzyme classes with potential roles in community metabolism. This is consistent with the proposed role of numerous uncharacterized microbial genes in explaining the vast majority of microbial diversity and function within the human gut[40]. This suggests that a concerted approach integrating computational function prediction with microbial physiological and biochemical validation will be necessary to link specific microbial chemistry to new individual organisms, genes, and enzymes from meta'omic sources.

**Predicted metabolites reveal global structure in the IBD metabolome.** Several recent studies demonstrated that IBD patients and healthy individuals, as well as the IBD subtypes (UC and CD), can be distinguished using metabolic profiling[41], suggesting that the IBD metabolome would be a meaningful benchmark for testing the accuracy of MelonnPan's metabolite predictions. To demonstrate that MelonnPan can capture biological variation in metabolic profiles without directly measuring the metabolites, we compared the first two components of metabolic variation from the measured metabolites and superimposed the predicted variation from inferred metabolites in the same two-dimensional space. Specifically, we ordinated the principal coordinates of 65 subjects in the NLIBD cohort based on Spearman dissimilarity between predicted and measured metabolite compositions for the top 50 unique metabolite clusters whose identities were confirmed against laboratory standards. The ordination plot revealed similar global structure in the IBD microbiome, which is reflected by the closeness of measured and predicted profiles across compounds (Fig. 4).

We found that well-predicted metabolites linked to IBD spanned a broad range of metabolic categories, including amino acids, bile acids, fatty acids, and sphingolipids, among others. In

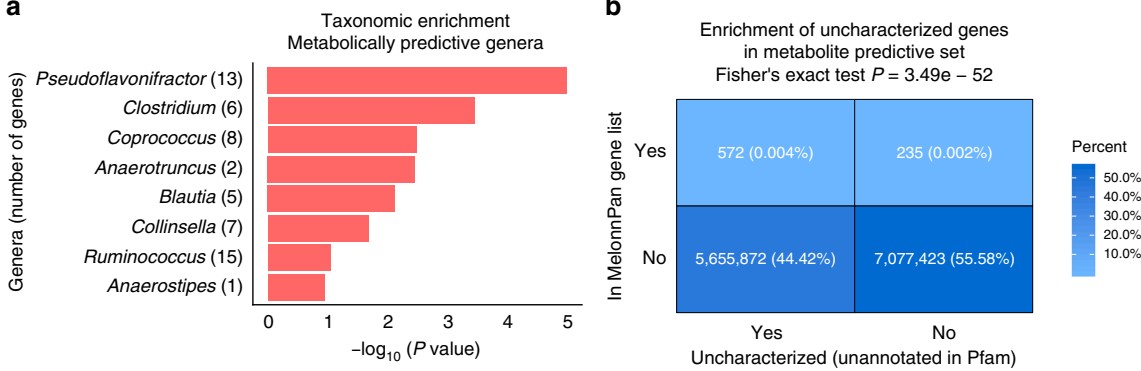

**Fig. 3** MelonnPan reveals biologically meaningful functional relationships. **a** Statistically significant gene sets (genera) ($Q < 0.25$) enriched in the MelonnPan predictive gene list, as identified by the permutation-based Kolmogorov–Smirnov (KS) test (based on 100,000 null permutations). The bars in the *x* axis indicate the logarithm of *P* values calculated as the fraction of permutation values that are at least as extreme as the original KS statistic derived from the non-permuted data. Numbers in the parentheses indicate the size of the gene sets. **b** Statistically significant over-representation of uncharacterized gene families in MelonnPan gene set. Contingency table describing the relationship between class membership in Pfam database and metabolite predictiveness reveals enrichment of uncharacterized proteins in the metabolite prediction process

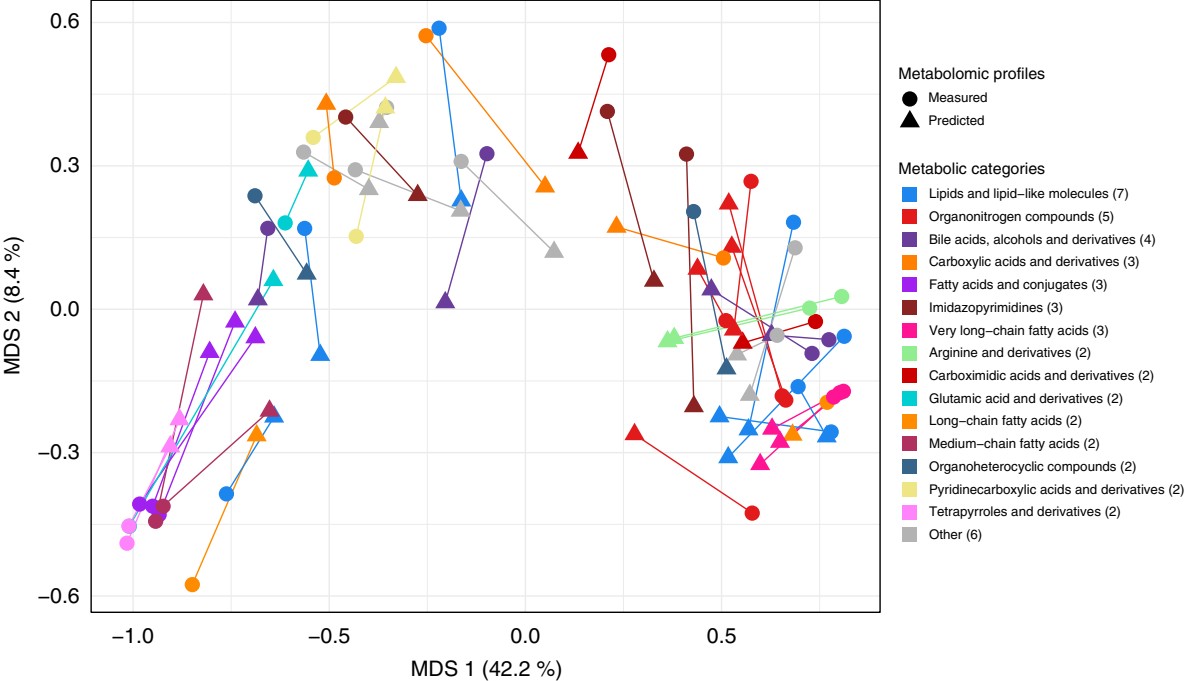

**Fig. 4** Predicted and measured metabolite profiles reveal similar global structure in the inflammatory bowel disease (IBD) microbiome. Principal coordinate analysis (PCoA) of top 50 well-predicted unique metabolite clusters whose identities were confirmed against laboratory standards, using a Spearman distance matrix. Metabolites in the PCoA plot are coloured by their labels and shaped by whether they were measured or predicted, and for each metabolite, the measured and predicted abundances are connected by edges, demonstrating closeness of measured and predicted metabolites across 65 Netherlands IBD (NLIBD) samples in the two-dimensional multidimensional scaling (MDS) space

particular, the ordination revealed strong support for approximately three clusters in the IBD metabolomic structure. Major ordination groups of covarying metabolites included compounds that are either derived from the same parent compound or interconverted by a common pathway, including: (i) several bile acid and very long-chain fatty acid groups depleted in IBD (Supplementary Fig. 11, right cluster), (ii) several cholesterol and tetrapyrroles derivatives enriched in IBD (left cluster), and (iii) a smaller mix of non-differentially abundant metabolites such as amino acids, peptides, purines, and their derivatives (central). Using the same differential abundance analysis ("Methods") on both predicted and measured metabolomic compositions yielded highly similar quantitative results across metabolites (Spearman correlation between effect size estimates based on measured and predicted profiles = 0.70 for CD versus HC and 0.45 for UC versus HC comparisons, respectively; $P < 2.2e-06$; Supplementary Fig. 12), suggesting that MelonnPan predictions can be used to infer disease-relevant differences in metabolomic compositions from metagenomes even in the absence of comprehensive metabolomic profiling.

We next set out to identify broad classes of compounds that were significantly over- or under-abundant in MelonnPan predictions for IBD. We focused on classes with at least one member in our data set and identified enrichments in MelonnPan prediction that were statistically significant after FDR correction (Fisher's exact test $Q < 0.25$, "Methods"). Two metabolic classes were significantly over-abundant in MelonnPan-predicted compounds, with the strongest effects observed among bile acids and tetrapyrroles (Supplementary Fig. 13). The enrichment of bile acid-associated products among the well-predicted metabolites highlights the important role of community ecology in microbial metabolism of bile acids. Concordant with previous studies, primary bile acids were significantly elevated and secondary bile acids were significantly reduced in IBD patients[31]. Bile acid

biosynthesis is directly mediated by microbial enzymatic activity, which, in IBD, fails to de-conjugate primary bile acid, causing a reduction in secondary bile acids and their anti-inflammatory effects on intestinal epithelial cells. Among other enriched metabolic classes, tetrapyrroles tended to be consistently depleted in IBD subjects compared to controls[31]. Taken together, these findings confirm that MelonnPan is able to provide metabolically relevant predictions across a broad range of compounds, identifying important contributors of the gut microbiome–metabolome axis, in turn facilitating large-scale integrated multi'omic analyses of the microbiome.

**Inference across the human body and environmental microbiomes.** As a final illustration of MelonnPan's ability to generate biological insights from a variety of microbial environments and assay types, we applied MelonnPan to three paired 16S and metabolomics datasets. These data were previously generated in the context of (i) metabolomic and taxonomic profiles from microbial communities associated with ecologically critical reef-building corals[42], (ii) bacterial community and metabolomic profiles from the vaginal microbiome[43], and (iii) paired taxonomic and metabolomic samples from the mouse gut[44]. Samples from each of these data sets were profiled for taxonomic composition by 16S rRNA gene amplicon (16S) sequencing. For metabolites, proton-nuclear magnetic resonance ($^1$H-NMR) spectroscopy was used for Data set 1, whereas targeted LC-MS and a combination of untargeted LC-MS and gas chromatography (GC)-MS metabolomics techniques were used for Data sets 2 and 3, respectively ("Methods").

In each of these datasets, we used MelonnPan to learn a model that predicted the relevant metabolite relative abundance features from available microbial features (taxonomic profiles derived from 16S rRNA gene sequencing). Owing to the small sample size

of these data sets, we used leave-one-out cross-validation (LOOCV) for MelonnPan training, followed by independent filtering (i.e. features were removed when they did not vary in value over the available samples) of both metabolite and operational taxonomic unit (OTU) features ("Methods"). Of the metabolites assayed in each data set, >50% did not pass individual pre-filtering (see "Methods", Supplementary Table 2) and were accordingly discarded from downstream analysis. We found that >60% of the analysed metabolites were well predicted in each of these data sets (Supplementary Fig. 14, Supplementary Table 2), and these were typically associated with a small number of OTUs (median model size of 29, 14, and 32 for Data sets 1–3, respectively), suggesting that, for a substantial fraction of compounds, information contained in a few taxonomic features is sufficient to explain a majority of the variation in metabolite abundances (in agreement with the human gut gene family application). Once again, the data-derived models learned by MelonnPan were significantly more accurate than the mechanistic models used by MIMOSA (Supplementary Table 2), which were limited to a very small number of already-characterized compounds in these more challenging microbial environments (Supplementary Figs. 15–17), further emphasizing the utility of MelonnPan for microbe–metabolite hypothesis generation in a variety of ecological settings. As a cautionary note, unlike the human gut samples, we did not have access to independent validation data sets in these environments. Therefore, we consider these applications as only a preliminary evaluation about the potential generalizability (external validity) of MelonnPan predictions in less well-studied environments.

## Discussion

MelonnPan represents a newly developed method to infer approximate metabolite feature abundances associated with microbial communities, and its validation and applications show that the information contained in microbiome taxonomic and functional profiles is sufficiently correlated with metabolomic content to infer actionable predictions of microbial community biochemical environments. This is of particular interest not only in the human gut but also generalized to a broad range of habitats including environmental microbiomes, given sufficient training measurements from the environments of interest, and the model provides an estimate of expected performance (the RTSI score) in new samples to guide experimenters. Although MelonnPan's predictive approach does not replace metabolomic profiling, it can approximately predict and compare possible metabolic profiles across many samples at a small fraction of the cost of metabolomics, thus opening up avenues for more cost-effective tiered study designs and providing metabolic insights and hypothesis generation in thousands of existing samples for which only metagenomic data are currently available.

In order to guide users when integrating MelonnPan hypotheses with downstream experimental validation, MelonnPan specifically provides a confidence score (RTSI) for each new microbiome, with a low confidence score indicating a high degree of dissimilarity with training metagenomes. Because training dissimilarity among metagenomes (as captured by RTSI) affects MelonnPan accuracy, RTSI values can be used as a guideline to indicate how much additional metabolomic data may be needed to complement a pre-trained MelonnPan model in a new environment. This information is particularly crucial since MelonnPan captures metagenome–metabolome associations in a data-driven manner, operating even in the absence of any microbial biochemical annotations, and this yields significantly higher prediction accuracy than current methods that rely on the very

limited number of well-characterized enzyme–metabolite relationships.

As with many recent studies, this investigation supports the importance of characterizing microbial features at the highest possible taxonomic resolution, as major microbial phenotypic differences associated with secondary metabolite production are often species or strain specific. Specifically, our analysis confirmed several previously documented IBD-associated species as important drivers of microbe–metabolite dynamics in the gut (Fig. 3a, Supplementary Data 7). To further interrogate whether species abundance data lead to similar metabolic prediction in the human gut application, we performed additional analysis with the species abundance data as input predictors to the MelonnPan model. While species-level predictors led to similar performance in the training cohort, these taxonomy-based predictions did not generalize to the independent cohort (Supplementary Fig. 18). This substantially lower predictability in the validation cohort likely reflects strain-level effects captured by gene family data, as strain differences in different populations can substantially affect metabolite prediction generalizability. This highlights the importance of including gene-level profiles as predictors, as specific strain-specific metabolism as well as other phenotypically relevant traits (e.g. antibiotic resistance) may not be captured from species abundance data alone.

The limitations of this approach must be considered in interpreting MelonnPan predictions. MelonnPan does not predict metabolite fluxes or peaks directly (as opposed to constraint-based methods); instead, it provides an estimate of each metabolite's community-wide relative abundances by synthesizing and combining microbial sequence features. The initial applications shown here are primarily applied to untargeted MS-based metabolomic measurements, but we have shown that MelonnPan remains comparably accurate when learning from targeted MS or NMR metabolite measurements as well. Although MelonnPan can be used for predictions in a broad range of environments beyond the well-studied human gut, it should be cautioned that each learned model is environment specific. Thus a model learned on the human gut can generalize to other human gut phenotypes (Supplementary Fig. 19), but no single model is expected to be accurate for cross-environment prediction tasks. MelonnPan is thus intended as a hypothesis-generation tool, as the general agreement between predicted and measured metabolite relative abundances is often sufficient to inform subsequent experimental validation studies, which should absolutely be performed to confirm predictions and obtain direct measurements of the metabolites of interest.

Interestingly, even for predictions made independently of mechanism and molecular origin, the strong predictability of some specific metabolites may have value in suggesting such mechanisms. For instance, predictability of a metabolite could indicate that it is either produced by a set of microbes or stimulated in host cells in the presence of specific microbes. Such specialized metabolites would be of use as markers of a defined set of microbes, a route for model-based association discovery that is faster and less resource-intensive than approaches such as Flux Balance Analysis. Future work identifying from culture-independent population-level data thus have the potential to focus on strain-specific gene sets or potentially even single-nucleotide polymorphism-level differences among bioactive taxa. Additional directions for future research to further refine the predictive accuracy of MelonnPan include (i) integration of other types of microbial measurements such as metatranscriptomic data, (ii) dynamic prediction utilizing longitudinal profiles, and (iii) adoption of more sophisticated machine learning strategies such as a multivariate or a Bayesian framework, which could explicitly incorporate quantitative features such as community-

wide enzyme-specific reaction information and zero-inflation, among others.

Culture-independent metagenomic sequencing has already profiled tens of thousands of samples containing millions of taxa and microbial genes[45–48]—millions of which are, as a result, uncharacterized. For example, only about 1.0% of all proteins in UniProtKB have been experimentally characterized[49]. As a result, one important finding of MelonnPan's human gut model is the association of mostly (>60%) unannotated gene families with metabolite relative abundances (Supplementary Data 7). Such links between gene families and metabolites provide promising targets for downstream characterization of the genes themselves, particularly when applied to other less well-characterized environments, as they may act functionally in the generation or metabolism of these compounds. This computational approach thus (i) generates both biochemical and functional genomic hypotheses for future validation, (ii) contributes to a system-wide understanding of the microbiome[50], (iii) serves as an additional, complementary tool to existing metabolic reconstruction models, and (iv) helps to lay the experimental design foundation for translational applications of metabolomics in microbial communities. As reference databases that allow matching to known standards continue to saturate[51,52] and training data sets continue to expand[53], the prediction accuracy of MelonnPan will improve by default over time. Taken together, this analytical framework is a necessary first step towards population-level meta'omic data integration, ultimately allowing us to better understand the dynamics of the microbiome, moving beyond molecular catalogues towards health applications of microbiome research.

## Methods

**Training and validation cohort descriptions.** Both training and validation cohorts are described in detail in Franzosa et al.[31]. Briefly, subjects included in the training cohort are from PRISM, which is a referral centre-based, prospective cohort. Patients aged ≥18 years with a diagnosis of CD based upon standard endoscopic, radiographic, and histologic criteria were eligible to participate. A total of 155 adult patients comprising of CD and UC patients and non-IBD controls (68 CD, 53 UC, 34 HC subjects) were enrolled[31]. PRISM research protocols were reviewed and approved by the Partners Human Research Committee (#2004-P-001067), and all experiments adhered to the regulations of this review board.

Subjects included in the validation cohort are from two independent cohorts from the Netherlands. Cohort 1 consists of 22 non-IBD (HC) subjects who participated in the general population study LifeLines-DEEP (LLDeep) in the northern Netherlands[54] and Cohort 2 consists of 43 patients with IBD (UC = 23, CD = 20) from the Department of Gastroenterology and Hepatology, University Medical Center Groningen (UMCG), Netherlands. A total of 65 stool samples were collected. Identical protocols were used to collect the stool samples in both these cohorts.

**Taxonomic and functional profiling.** Metagenomic data generation and processing were performed at the Broad Institute. After extracting DNA from stool samples, metagenomic libraries were prepared using the Nextera XT DNA Library Preparation Kit (Illumina) according to the manufacturer's recommended protocol and sequenced on the Illumina HiSeq 2500 platform, targeting ~2.5 Gb of sequence per sample with 101 bp, paired-end reads. Low-quality reads <60 nt in length were filtered out using Trimmomatic[55] as well as human contaminating reads using bowtie2[56]; these steps were performed using the KneadData pipeline (https://bitbucket.org/biobakery/kneaddata). Species-level taxonomic abundances were inferred for all samples using MetaPhlAn2[57] (https://bitbucket.org/biobakery/metaphlan2) and run with default parameters. Functional profiling was performed by using HUMAnN2[29]. Briefly, HUMAnN2 maps metagenomic reads to the pangenomes[58] of species identified upstream in the taxonomic profiling step. Protein-coding sequences in these pangenomes have been pre-annotated to their respective UniRef90 families[30], which serve as a comprehensive, non-redundant protein sequence database. Reads that do not align to a known pangenome are separately mapped to the entirety of UniRef90 by translated search with DIA-MOND[59]. All hits are weighted based on alignment quality and sequence length, with per-species and unclassified hits combined to produce community totals for each protein family (in addition to species-stratified totals) in RPK (reads per kilobase) units. RPK units were further normalized to RPKM units (reads per kilobase per million sample reads) to account for variation in sequence depth across samples.

**Metabolite profiling.** The stool samples from participants in the two cohorts (PRISM cross-sectional, 155 samples and NLIBD, 65 samples; weight range 50.5–167.8 mg) were processed as described in Franzosa et al.[31]. Four separate LC-MS methods that measure complementary metabolite classes were used to measure polar metabolites and lipids in each sample. Raw LC-MS data were processed using Genedata Expressionist v9.0 for chemical noise removal, RT alignment, peak detection, and isotope clustering. The combination of the four LC-MS methods generated 8869 clustered features, characterized by chromatographic retention time and exact mass to <5 ppm accuracy. Three thousand eight hundred and twenty-nine metabolomic features were linked to putative identifiers based on accurate m/z matching against the HMDB[33]. A subset of 466 metabolites were identified more precisely using reference data generated from an in-house compound library. More details of the LC-MS metabolomics experiments are provided in Franzosa et al.[31].

**Filtering, transformation, and normalization.** After normalizing the raw measures into relative abundances, we limit our analysis to only those features (species, gene families, and metabolites) that are both prevalent and abundant with mean relative abundance >0.01% in at least 10% of the samples. Because of the specific properties of meta'omic data that significantly influence model building, such as compositionality, sparsity, skewness, mean-variance dependency, and extreme values, we quantile-transform the input features (species or gene family abundances) to the quantiles of a standard normal distribution in order to improve the detection power of the elastic net model[60,61]. This approach has been extensively used in genetic association studies and have proven to be a robust approach for modelling non-normal phenotypes[62–64]. We use the *rntransform* function from the *GENABEL* package from R (version 3.5.1) for the quantile-based inverse normal transformation[65]. To identify an optimal subset of predictive features, the metabolite relative abundances are arcsine square root transformed[66] to approximate homoscedasticity when applying linear models. The models are fitted to the transformed data and the resulting predictions are back-transformed to preserve the coverage of the predicted metabolite compositions.

**Elastic net regularization.** We designed and implemented the elastic net regularization technique[28] for metabolomic predictive model building. In particular, a per-metabolite elastic net model is fit on the rank-transformed features (species or gene family abundances). More details on the elastic net method and its variants have been previously published[67]. We use the *glmnet* package from R (version 3.5.1) for fitting the elastic net model and choose the tuning parameters (i.e. both the elastic net mixing parameter $\alpha$ and sparsity parameter $\lambda$) based on cross-validation.

**Cross-validation and evaluation metric.** Ten-fold cross validation (unless otherwise stated) was used to determine the tuning parameters in the elastic net model. Spearman correlation coefficient ($r$) between the true and predicted metabolite compositions was used to evaluate the predictability of each compound. Following Cohen[68], we term those metabolites for which $r$ is >0.3 as "well predicted" and flag the rest as poorly predicted metabolites.

**Significance testing with shuffled data.** In order to quantify whether our framework identified more well-predicted metabolites than expected by chance (i.e. when all the shared signal between genes and metabolites are broken), we repeatedly shuffled the sample labels in both metabolite and gene family tables, applied the MelonnPan model using the randomized data to link genes to metabolites, and compared the number of well-predicted metabolites obtained with these randomized data to the number of well-predicted metabolites obtained with the original data. Random data were generated following the approach outlined in the R (version 3.5.1) package *pecante* using the function *randomizeMatrix*, which employs a post-permutation renormalization within sample to preserve the core structural characteristics of the original data set. The procedure was repeated 1000 times to estimate the null distribution of the prediction performance in both the training and validation cohorts.

**RTSI score.** To calculate the RTSI for new samples, we employed PCA to extract the continuous axes of variation that reflect the population structure in the training metagenomes, following a quantile-based inverse normal transformation. Specifically, we selected top PCs based on the Tracy–Widom statistics[69]. Based on the top PCs, we classified the NLIBD samples as either similar or dissimilar based on their highest correlation with the extracted PCs. We refer to the resulting similarity (correlation) as the RTSI score. We used the *AssocTests* package from R (version 3.5.1) and a significance threshold of 0.05 to select the number of top PCs.

**Gene set enrichment analysis.** We conducted two types of enrichment analysis after constructing appropriate gene sets: over-representation analysis and GSEA. For the over-representation analysis, we created two-by-two tables comparing the number of candidate genes that are members of the category to those that are not members of the category and assessed the significance of over-representation using a one-tailed Fisher's exact test. For the GSEA[35], we calculated enrichment score based on a KS test statistic that reflects the degree to which a gene set is over-represented at the

extremes (top or bottom) of the entire ranked list of genes (ranking gene family features by their overall predictability in the metabolite features). To assess significance, we performed 100,000 permutations for each gene set using the functionality of the R (version 3.5.1) package gsEasy. For both these analyses, we corrected for multiple hypothesis testing using a Benjamini–Hochberg false discovery rate (FDR) approach[70].

**Differential abundance analysis**. To perform differential abundance analysis of the measured and predicted metabolomics data in the NLIBD cohort, we fitted a linear model to each log-transformed metabolite relative abundance profile separately, after adjusting for disease status (with HC as reference category), age, and medications: immunosuppressants (yes/no) and anti-inflammatory (yes/no).

**Non-gut and non-human microbial profiles**. We obtained several previously published data sets from publicly available databases for non-human-gut environments, each pairing 16S rRNA gene-based taxonomic data with metabolomic profiles[24,42]. In each of these data sets, we utilized the processed profiles, publicly available through the authors, but the relevant sequence data are also available through NCBI. For Data set 1, we used taxonomic profiling provided from 16S rRNA gene sequencing coupled with [1]H-NMR-based metabolomics[42]. For vaginal samples in Data set 2, 16S rRNA gene analysis was performed from vaginal swabs, and paired cervicovaginal lavage fluid was collected for metabolomic analysis (targeted LC-MS for 180 compounds[24,43]. In Data set 3, taxonomic composition was again assayed using 16S rRNA gene amplicon sequencing, and metabolites were measured using global LC- and GC-MS metabolomics[24,44]. Both OTU and metabolite features with <0.0001% relative abundance in >10% of samples were discarded from downstream analysis. In addition, a variance filtering step was applied to remove features with very low variance. MelonnPan's elastic net regularization with an LOOCV was then applied to the quality-controlled profiles.

**Reporting summary**. Further information on research design is available in the Nature Research Reporting Summary linked to this article.

## Data availability
Metagenomic sequences for the PRISM, LLDeep, and NLIBD cohorts are available via SRA with BioProject number PRJNA400072. PRISM metabolomics data (accession number PR000677) are available at the NIH Common Fund's Metabolomics Data Repository and Coordinating Center (supported by NIH grant, U01-DK097430): Metabolomics Workbench (http://www.metabolomicsworkbench.org). Amplicon data are available as cited from original publications. A pre-trained model using HUMAnN2-derived UniRef90 gene family features from the human gut is included as part of the MelonnPan software package.

## Code availability
The implementation of MelonnPan is publicly available with source code, documentation, tutorial data, and as an R package at http://huttenhower.sph.harvard.edu/melonnpan. The software packages used in this work are free and open source, including bioBakery methods available via http://huttenhower.sph.harvard.edu/biobakery as source code, cloud-compatible images, and installable packages. Analysis scripts using these packages to generate figures and results from this manuscript (and associated usage notes) are available from the authors upon request.

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

## Acknowledgements

We thank Tiffany Poon for project management and Heather Kang for editorial assistance and feedback on the manuscript. This work was funded in part by the US National Science Foundation (NSF CAREER DBI-1053486 and NSF EAGER MCB-1453942 to C.H.) and the National Institutes of Health grants R01HG005220 (to C.H. and Rafael Irizarry), U54DK102557 (to C.H. and R.J.X.), R01DK92405 (to R.J.X.), R24DK110499 (to C.H.), and Crohn's & Colitis Foundation of America (CCFA) award #3162, and the Center for Microbiome Informatics and Therapeutics (6933665 PO no. 5710004058 to R.J.X.).

## Author contributions

H.M., E.A.F., H.V., R.J.X. and C.H. conceived the study. H.M., L.J.M., S.B., A.S.-M., A.D.K. and C.B.C. generated the data and performed the analysis. H.M., E.A.F., H.V. and C.H. wrote the paper.

## Additional information

**Competing interests:** R.J.X. is a consultant to Novartis and Nestle. The other authors declare no competing interests.

