## [Peer Review File · Nature Communications]

Reviewers' comments:

Reviewer #3 (Remarks to the Author):

The manuscript has undergone several rounds of reviews and the authors have improved the text: (1) rewriting the introduction, methods, and discussion in a way that is more clear for readers from a diverse background; (2) expressing the random sampling as averages of repeated iterations, instead of reporting a single realization of their null distribution and; (3) including predictions for coral, murine, and vaginal samples, suggesting that the method can be used for diverse environments.

[Redacted] I find the manuscript scientifically sound and of relevance for the fields of microbiome and metabolomics. The major remaining drawback is that specific microbiome/metabolomics training data, derived from the environment of interest, is needed to be able to make predictions about other samples from that same environment. In the rebuttal they state: "To clarify, the creation of a trained prediction model depends on the availability of at least one paired microbiome/metabolome dataset from an environment comparable to that in which the model is intended to be applied." While in the gut samples training and testing data was separated, the coral, murine, and vaginal models were trained on the same datasets as were predicted. As it is expected from supervised regression models such as elastic nets, they could "predict" back about half of the features learned from their trained weights, but this may say little about how well the model could be generalized for predicting metabolites in these environments.

I am satisfied with the new random tests that were performed [Redacted] and expressed as an average instead of a single random point and also with the explanation about the relative metabolome abundance.

For more clarity I suggest reserving the term "well-predicted" for the samples that were predicted independently of the training sets such as the validation set of the human gut samples, while the samples from the training set that exhibit a moderate or high correlation in the models that they were trained could be called "well-correlated" or another term that highlights the difference between using the weight matrix to predict independent samples and finding high correlation during the training.

As a suggestion to improve the interpretability of the RTSI score for a broader audience, I recommend considering simpler measures such as the average similarity of the training sets and the test sets or the percentage of coverage of the non-zero features of the trained model and the test samples to evaluate if they would perform better or equally well compared to the current RTSI score (which does not perform really great) with the additional benefit of providing an intuitive explanation for how well can a sample be represented by a given training set.

I understood that the authors used Cohens empirical rule of thumb regarding the effect-size in social science to define the 0.3 rank correlation cutoff. I insist, though that the references are not really appropriate since it is enough to cite the original source (Cohen 1988) and as with the common criticism for the usage of such a "rule-of-thumb" (as mentioned in the other reference provided by the authors, Nakagawa & Cuthill 2007), there is no indication that the data shows such critical point, conversely it is fairly linear at 0.3. With this said, as long as it is presented as such, there is nothing particularly wrong in defining an arbitrary cutoff as a choice to make predictions for a sufficiently high number of metabolites without compromising too much on the performance.

Response to Reviewers' Comments

The manuscript has undergone several rounds of reviews and the authors have improved the text: (1) rewriting the introduction, methods, and discussion in a way that is more clear for readers from a diverse background; (2) expressing the random sampling as averages of repeated iterations, instead of reporting a single realization of their null distribution and;(3) including predictions for coral, murine, and vaginal samples, suggesting that the method can be used for diverse environments.

[Redacted] I find the manuscript scientifically sound and of relevance for the fields of microbiome and metabolomics.

Reviewer 3, Response 1: We thank Reviewer 3 for this very positive evaluation of the manuscript.

The major remaining drawback is that specific microbiome/metabolomics training data, derived from the environment of interest, is needed to be able to make predictions about other samples from that same environment.

In the rebuttal they state: "To clarify, the creation of a trained prediction model depends on the availability of at least one paired microbiome/metabolome dataset from an environment comparable to that in which the model is intended to be applied."

While in the gut samples training and testing data was separated, the coral, murine, and vaginal models were trained on the same datasets as were predicted. As it is expected from supervised regression models such as elastic nets, they could "predict" back about half of the features learned from their trained weights, but this may says little about how well the model could be generalized for predicting metabolites in these environments.

Reviewer 3, Response 2: First, many thanks for the reviewer's input. In agreement, we do not expect a MelonnPan model trained in one environment (e.g. human gut) to generalize to another environment with sufficiently distinct metabolic activities (e.g. coral). To this effect, we have now added a cautionary statement in the manuscript to indicate different degrees of expected generalization when interpreting MelonnPan results for environment-specific versus cross-environment predictions tasks:

'Although MelonnPan can be used for predictions in a broad range of environments beyond the well-studied human gut, it should be cautioned that each learned model is environment-specific. Thus a model learned on the human gut can generalize to other gut phenotypes (as shown above), but no single model is expected to be accurate for cross-environment prediction tasks.'

The reviewer's point regarding the generalizability of MelonnPan predictions in non-human-gut environments is an important one, which (as suggested by the revised text) we have now evaluated more thoroughly. First, to assess generalizability across human-specific phenotypes, we have now mimicked a cross-study analysis by creating three independent datasets consisting of CD, UC, and healthy control (HC) samples in the combined PRISM and NLIBD cohorts. We then independently trained MelonnPan models within each dataset. Finally, we used these individually cross-validated models to generate predictions on the holdout datasets. The results from this are shown below:

Cross-phenotype predictions in PRISM + NLIBD
No. of well-predicted metabolites: n (%)

Similar to our validation experiments in the manuscript, a substantial number of metabolites remained well-predicted across phenotypes during validation within and between studies. This is particularly apparent for diseased samples (i.e. CD and UC); but even in the difficult case of generalizing prediction results from healthy controls to IBD patients, a moderate number of metabolites remained well-predicted in both CD and UC validation samples (22% and 33% respectively).

Further, as expected and consistent with the prior studies (PubMed ID: 27400279), predictions transferred between disease groups were, in general, still biologically useful but overall less accurate than models within-study. This calls for the exercise of caution in using MelonnPan for cross-environment prediction tasks, which is true both across human phenotypes and between even more extreme environmental differences (e.g. human vs. non-human). This is not a limitation of the MelonnPan methodology *per se*, but instead a property of *any* machine learning algorithm: accuracy diminishes when an application dataset is dissimilar to the training dataset. This is exactly the reason we initially created the RTSI score, which provides quantitative guidance to users when an application dataset is “too dissimilar” for biological utility (in addition to the qualitative guidance added to the text above).

Taken together, these results thus indicate that an internally validated model is a quantifiably good indicator of generalizability, at least within the same environment. Our additional cross-study validation experiments further confirm this finding, indicating that the default MelonnPan models are sufficient to provide users a good sense of the expected MelonnPan performance in new samples from similar environments. The method provides both several tunable parameters as well as the RTSI scores as guidance to customize their analysis within a given setting or environment, and gives users an easy way to train models appropriate for new environments.

I am satisfied with the new random tests that were performed [Redacted] and expressed as an average instead of a single random point and also with the explanation about the relative metabolome abundance.

Reviewer 3, Response 3: Thank you.

For more clarity I suggest reserving the term "well-predicted" for the samples that were predicted independently of the training sets such as the validation set of the human gut samples, while the samples from the training set that exhibit a moderate or high correlation in the models that they were trained could be called "well-correlated" or another term that highlights the difference between using the weight matrix to predict independent samples and finding high correlation during the training.

Reviewer 3, Response 4: As rightly mentioned by the reviewer, it is important to separate internal and external validity of a prediction model, which correspond to two distinct aspects of prediction performance of any machine learner. While internal validity of a prediction model refers to the validity of claims for the underlying population that the data originated from, external validity refers to the generalizability of claims to 'plausibly related' populations. We acknowledge that external validation is commonly considered a stronger test for prediction models than internal validation, as it addresses transportability rather than reproducibility and agree that distinguishing these two sets of prediction tasks will avoid confusion for the readers.

For the interest of clarity, however, we respectfully disagree with the reviewer on the suggestion of using two different terms to distinguish individual features that are 'well-predicted in training' vs. 'well-predicted in validation'. Since the definition of 'well-predicted' is central to the MelonnPan methodology regardless of the availability of validation datasets, we believe that using two terminologies for "well-predicted" will further confuse the readers. We note that the previous publication on this topic (PMID: 27239563) also used the same terminology to call out specific "well-predicted" metabolites, and that this is standard in essentially any machine learning (within or even outside of biological applications). Therefore, with the interest of remaining consistent both with the literature and within the manuscript, we have opted to use the same terminology as before with added cautionary note when applicable (e.g. in non-human-gut samples), as shown in the revised manuscript as follows:

'As a cautionary note, unlike the human gut samples, we did not have access to independent validation datasets in these environments. Therefore, we consider these applications as only a preliminary evaluation about the potential generalizability (external validity) of MelonnPan predictions in less well-studied environments.'

As a suggestion to improve the interpretability of the RTSI score for a broader audience, I recommend considering simpler measures such as the average similarity of the training sets and the test sets or the percentage of coverage of the non-zero features of the trained model and the test samples to evaluate if they would perform better or equally well compared to the current RTSI score (which does not perform really great) with the additional benefit of providing an intuitive explanation for how well can a sample be represented by a given training set.

Reviewer 3, Response 5: This is an excellent suggestion. In agreement, in this round of revision, we have extensively investigated the utility of several simple measures, which are arguably easy to interpret. We ultimately revised the RTSI score to use a simpler Principal Component Analysis (PCA) based similarity, which is now the default in MelonnPan's implementation and updated throughout the manuscript.

During the associated evaluation, we also considered two distance-based measures as a proxy for average similarity, as suggested by the reviewer. We detail the relevant results from this effort below. Specifically, these include (i) median Bray-Curtis dissimilarity from training samples (**Panel A**) and (ii) median Euclidean distance from training samples, followed by a quantile-based inverse normal transformation (**Panel B**). We additionally considered two simplistic measures based on the feature distributions. These include (i) a Kolmogorov Smirnov (KS) test-based approach (**Panel C**) in which, two-sample KS tests are sequentially applied to each test sample to determine its similarity (or dissimilarity) with the training samples and the resulting p-values are combined using Fisher's method, producing a single Chi-square statistic (i.e. average dissimilarity) per test sample, and (ii) a coverage-based approach as suggested by the reviewer (**Panel D**), in which we simply calculate the percentage of coverage of the top 100 predictive features in test samples. While all these measures are easy to interpret, they did not correlate well with the MelonnPan prediction accuracy in NLIBD validation samples, which also remain consistent within a 'reasonable' variation of the above approaches (e.g. average instead of median and vice versa or top 200 or higher number instead of 100).

As was ultimately used as noted above, we further employed a PCA-based approach. Our formulation is motivated by the widespread use of PCA in human genetics, where the top PCs are viewed as continuous axes of variation that reflect population structure. To this effect, we extracted the top PCs explaining a majority of variation in the training metagenomes and calculated the average similarity (correlation) by sequentially seeking the highest correlation between test samples and the extracted principal components. As shown in the plot below (new Supplemental **Fig. S10**), this approach provides modest correlation with MelonnPan prediction accuracy in the NLIBD samples with the additional benefit of providing an intuitive explanation of how well can a new sample be represented by a given training set. Based on this, we have adopted the PCA-based scores as our new RTSI scores.

The relevant text in the manuscript is updated as follows:

'Briefly, MelonnPan first flags any gene family not present in the training metagenomes, and for the remaining common gene families (between training and test samples), it calculates an average similarity score (per microbial community sample) based on Principal Component Analysis (PCA). In particular, RTSI scores are computed by sequentially seeking the highest correlation coefficient with the top PCs explaining a majority of the variation in the training metagenomes (**Methods**). We compared the RTSI scores and MelonnPan accuracies for all of the NLIBD validation samples across all well-predicted metabolites (**Fig. S10**), which revealed strong and statistically significant correlation (Spearman correlation = 0.4, P = 0.003) between representativeness (higher RTSI) and predictability of samples across metabolites (as measured by Spearman correlation between measured and predicted metabolite abundances). This correlation value was itself conservatively low, caused by a few outlier samples without which the predictiveness of the RTSI for MelonnPan performance on new samples is even greater.'

And the corresponding section in **Methods** now reads:

'To calculate the Representative Training Sample Index (RTSI) for new samples, we employed PCA to extract the continuous axes of variation that reflect the population structure in the training metagenomes, following a quantile-based inverse normal transformation. Specifically, we selected top PCs based on the Tracy–Widom statistics (71). Based on the top PCs, we classified the NLIBD samples as either similar or dissimilar based on their highest correlation with the extracted principal components. We refer to the resulting similarity (correlation) as the RTSI score. We used the *AssocTests* package from R (version 3.5.1) and a significance threshold of 0.05 to select the number of top PCs.'

I understood that the authors used Cohens empirical rule of thumb regarding the effect-size in social science to define the 0.3 rank correlation cutoff. I insist, though that the references are not really appropriate since it is enough to cite the original source (Cohen 1988) and as with the common criticism for the usage of such a "rule-of-thumb" (as mentioned in the other reference provided by the authors, Nakagawa & Cuthill 2007), there is no indication that the data shows such critical point, conversely it is fairly linear at 0.3. With this said, as long as it is presented as such, there is nothing particularly wrong in defining an arbitrary cutoff as a choice to make predictions for a sufficiently high number of metabolites without compromising too much on the performance.

Reviewer 3, Response 6: We thank the reviewer for the comment. In agreement, we have retained the same cutoff as before and only the Cohen reference, as suggested by the reviewer. To this effect, we have modified the relevant sentence in the text as follows:

'This represents both an indication of MelonnPan's robustness to overfitting and a justification for its arbitrary threshold, which is broadly consistent for summarizing a sufficiently high number of "well-predicted" metabolites as discussed above.'

REVIEWERS' COMMENTS:

Reviewer #3 (Remarks to the Author):

The authors have addressed our concerns.

I would like the caveat of having metabolomic training data available from similar microbial systems to be mentioned in the Abstract, but I will leave that decision to the discretion of the editor.

Reviewer #3 (Remarks to the Author):

The authors have addressed our concerns.

I would like the caveat of having metabolomic training data available from similar microbial systems to be mentioned in the Abstract, but I will leave that decision to the discretion of the editor.

We thank the reviewer for the comment. We have now included this caveat in the Abstract.